# Killer Immunoglobulin-Like Receptor 2DS2 (KIR2DS2), KIR2DL2-HLA-C1, and KIR2DL3 as Genetic Markers for Stratifying the Risk of Cytomegalovirus Infection in Kidney Transplant Recipients

**DOI:** 10.3390/ijms20030546

**Published:** 2019-01-28

**Authors:** Dominika Deborska-Materkowska, Agnieszka Perkowska-Ptasinska, Anna Sadowska-Jakubowicz, Jolanta Gozdowska, Michał Ciszek, Joanna Pazik, Agata Ostaszewska, Maciej Kosieradzki, Jacek Nowak, Magdalena Durlik

**Affiliations:** 1Department of Transplantation Medicine, Nephrology, Internal Diseases, T. Orłowski Institute of Transplantation Medical University of Warsaw, 59 Nowogrodzka Street, 02-006 Warsaw, Poland; dominika.deborska-materkowska@wum.edu.pl (D.D.-M.); agnieszka.perkowska-ptasinska@wum.edu.pl (A.P.-P.); andziaa@wp.pl (A.S.-J.); jola-md@prokonto.pl (J.G.); jpazik@poczta.fm (J.P.); 2Department of Immunology, Transplantology, Internal Diseases, T. Orłowski Institute of Transplantation Medical University of Warsaw, 59 Nowogrodzka Street, 02-006 Warsaw, Poland; mciszek@onet.pl; 3Department of General and Transplant Surgery, T. Orłowski Institute of Transplantation Medical University of Warsaw, 59 Nowogrodzka Street, 02-006 Warsaw, Poland; ag.ostaszewska@gmail.com (A.O.); mpkosieradzki@gmail.com (M.K.); 4Department of Immunogenetics Institute of Hematology and Transfusion Medicine 14 Indira Gandhi Street, 02-776 Warsaw, Poland; jnowak@ihit.waw.pl

**Keywords:** killer-cell immunoglobulin-like receptor, cytomegalovirus, natural killer cell, human leukocyte antigen, lymphocytopenia, kidney transplantation

## Abstract

Infection with cytomegalovirus (CMV) remains a major problem in kidney transplant recipients, resulting in serious infectious complications and occasionally mortality. Accumulating evidence indicates that natural killer cell immunoglobulin-like receptors (KIRs) and their ligands affect the susceptibility to various diseases, including viral infections (e.g., CMV infection). We investigated whether KIR genes and their ligands affect the occurrence of CMV infection in a group of 138 kidney transplant recipients who were observed for 720 days posttransplantation. We typed the recipients for the presence of KIR genes (human leukocyte antigen C1 [HLA-C1], HLA-C2, HLA-A, HLA-B, and HLA-DR1) by polymerase chain reaction with sequence-specific primers. The multivariate analysis revealed that the lack of *KIR2DS2* (*p* = 0.035), the presence of *KIR2DL3* (*p* = 0.075), and the presence of *KIR2DL2*–HLA-C1 (*p* = 0.044) were risk factors for posttransplant CMV infection. We also found that a lower estimated glomerular filtration rate (*p* = 0.036), an earlier time of antiviral prophylaxis initiation (*p* = 0.025), lymphocytopenia (*p* = 0.012), and pretransplant serostatus (donor-positive/recipient-negative; *p* = 0.042) were independent risk factors for posttransplant CMV infection. In conclusion, our findings confirm that the KIR/HLA genotype plays a significant role in anti-CMV immunity and suggest the contribution of both environmental and genetic factors to the incidence of CMV infection after kidney transplantation.

## 1. Introduction

Despite the introduction of novel diagnostic and therapeutic modalities for the management of cytomegalovirus (CMV), it remains a significant cause of serious posttransplant complications, with high rates of mortality and graft loss [1]. Donor (D) and recipient (R) CMV serostatus is an important determinant of the risk of CMV infection posttransplantation. Although serostatus is useful, it does not allow us to precisely predict who is at risk for infection. Despite the implementation of prevention strategies that are based on pretransplant D/R serostatus, a high incidence of late-onset CMV has occurred in all studies that have evaluated universal prophylaxis. Apart from pretransplant serology, other risk factors for CMV infection include certain types of transplant (e.g., lung), shorter courses of prophylaxis, disease severity, higher levels of immunosuppression, transplant failure, allograft rejection, and the presence of immune reconstitution [2,3,4,5]. 

The immune response to CMV infection is complex. The role of both adaptive and innate responses has not been well established in transplant recipients. Natural killer (NK) cells are effector cells that constitute a crucial part of the innate immune system and provide the first-line of defense against and control of CMV infection [6]. The inhibition and activation of the NK cell response is modulated by signals that include the interplay between polymorphic killer-cell immunoglobulin-like receptors (KIRs) and their ligands, human leukocyte antigen (HLA) class I molecules. There are two types of KIR molecules (activating and inhibitory) that are expressed on NK cells and minor subpopulations of T cells. KIRs that have long cytoplasmic tails (killer-cell immunoglobulin-like receptor-2DL1 (*KIR2DL1*), *KIR2DL2*, *KIR2DL3*, *KIR2DL5*, *KIR3DL1*, *KIR3DL2*, and *KIR3DL3*) suppress effector functions. KIRs that have short intracytoplasmic tails (*KIR2DS1*, *KIR2DS2*, *KIR2DS3*, *KIR2DS4*, *KIR2DS5*, and *KIR3DS1*) activate lymphocytes through an associated DAP12 molecule [7,8,9]. *KIR2DL4* encodes a receptor that performs both inhibitory and activating functions. Inhibitory KIRs recognize HLA class I molecules on the surface of target cells. Activating KIRs have lower affinity for HLA, and their natural ligands are less well documented. There are two categories of KIR haplotypes. Group A haplotypes carry mostly inhibitory KIRs; only *KIR2DS4* and *KIR2DL4* are activating ones. B-type haplotypes are more variable and contain more than one activating KIR gene other than *KIR2DS4* [10]. KIR and HLA genes have been identified on two separate chromosomes (chromosomes 19 and 6, respectively) and, thus, are inherited independently. Therefore, an individual may lack the corresponding HLA ligands for KIRs. Depending on KIR and KIR ligand genotypes, people may differ substantially in their NK response. The lack of ligands for inhibitory KIRs and the presence of activating KIRs in the recipient have been associated with a protective effect on the rate of CMV infection after kidney transplantation [11,12].

The aim of the present study was to analyze the association between post-kidney transplant CMV infection and the recipient’s KIR genotype and evaluate other possible risk factors for the occurrence of CMV infection in this patient population.

## 2. Results

### 2.1. Clinical Characteristics of the Study Patients

One hundred twenty-eight study participants received an organ from a deceased donor, and eight participants received an organ from a living related donor. Cytomegalovirus infection occurred in 36.23% of the 138 patients during 720 days after kidney transplantation. The pretransplant D/R serostatus and baseline recipients’ characteristics are summarized in Table 1.

*KIR2DL4*, *KIR3DP1*, *KIR3DL2*, and *KIR3DL3* were present in all of the recipients. The frequencies of other KIR genes are presented in Table 2.

### 2.2. The Relation between the Cmv Infection Occurrence and Individual Kir Gene Frequencies (Univariate Analysis)

We analyzed whether individual KIR genes influence the incidence of CMV infection. The KIR gene distribution between the CMV+ and CMV− groups is presented in Figure 1. 

The *KIR2DS2* gene frequency was 73.6% in the CMV− group vs. 26.4% in the CMV+ group (*p* = 0.012). The *KIR2DL2* gene frequency was 73.2% in the CMV− group vs. 26.8% in the CMV+ group (*p* = 0.017). We found nearly absolute concordance between *KIR2DS2* and *KIR2DL2* with regard to their presence. Therefore, these two genes, with the exception of one individual, had identical frequencies. No significant difference was found in the other KIR gene frequencies between the CMV+ and CMV− groups (Figure 1).

### 2.3. KIR Genotype and Rate of Cmv Infection (Univariate Analysis)

Based on the presence or absence of multiple activating KIRs, we categorized the genotypes as AA (containing *KIR2DS4* as the only activating gene) or B/X type (carrying activating KIR genes other than *KIR2DS4*), which is in accordance with the published literature [10,13]. Forty-two (30.4%) individuals were assigned to the KIR A/A genotype; the remaining 96 (69.6%) individuals were assigned to the KIR B/X genotype. We noted a trend toward a lower incidence of CMV infection in recipients who carried the KIR B/X genotype (*n* = 30 [31.2%]) compared with the KIR A/A genotype (*n* = 20 [47.6%]). Nevertheless, the difference between both KIR genotypes did not reach statistical significance (*p* = 0.065; Table 3).

### 2.4. Cumulative Number of Activating and Inhibitory Genes That Were Present in Kir Haplotypes and Cmv Infection (Univariate Analysis)

We analyzed whether the cumulative number of either activating or inhibitory KIRs in the recipient’s genome varied between CMV+ and CMV− patients. We did not find such a difference (Figure 2a,b). 

### 2.5. KIR/HLA Interactions and CMV Infection (Univariate Analysis)

To determine whether certain receptor-ligand combinations influence the development of CMV infection, we calculated the number of matches for certain KIRs with known HLA ligands. Univariate analysis did not reveal significant associations between certain KIR/HLA matches and posttransplant CMV infection (Table 3). 

### 2.6. Factors Other Than KIR Factors That Affect Posttransplant Cmv Infection

Univariate logistic regression analysis was performed to examine factors that could potentially impact the posttransplant occurrence of CMV infection. Of these factors, allograft function (*p* = 0.04 for recipients with a lower GFR on day 30; *p* = 0.0006 for recipients with a lower GFR on day 90), lymphocytopenia before day 90 posttransplantation(*p* = 0.0003), leucopenia (*p* = 0.007), higher pretransplant recipient CMV IgG titer (*p* = 0.016), pretransplant D/R CMV serostatus (*p* = 0.019 for D+/R−), and an earlier time of antiviral prophylaxis initiation (*p* = 0.005) were statistically significant. All of other variables were not associated with CMV infection (Table 4).

### 2.7. Independent Risk Factors for Posttransplant Cmv Infection (Multivariate Analysis)

The multivariate analysis included all clinical variables and individual KIR genes and their HLA ligands that were analyzed in the univariate analyses. The maximum model to be considered was identified. The optimal subset of variables was then selected, and their reliability was assessed. *KIR2DL2* and *KIR2DS2* are in strong linkage disequilibrium and are expected to be expressed together. In the present study *KIR2DL2* and *KIR2DS2* were present in the same group of individuals, with the exception of one recipient. Therefore, statistical models could include either *KIR2DL2* or *KIR2DS2*. The multivariate analysis revealed the following independent risk factors for posttransplant CMV infection: lack of *KIR2DS2*, presence of *KIR2DL2-HLA-C1* and *KIR2DL3*, allograft dysfunction, earlier time of antiviral prophylaxis initiation, lymphocytopenia before day 90 posttransplantation, and high-risk D/R serostatus (Table 5).

A Kaplan Meier curve of the time to CMV infection in recipients with KIR2DS2 vs. recipients without KIR2DS2 is shown in Figure 3.

## 3. Discussion

The present study analyzed the association between KIR genotype diversity in kidney transplant recipients and the occurrence of CMV infection. The key finding of this study was the association between the lack of activating *KIR2DS2* and posttransplant CMV infection. This phenomenon was also described in patients after hematopoietic cell transplantation (HCT). In the study by Zaia et al., the presence of *KIR2DS2* and *KIR2DS4* in the donor genotype was associated with lower CMV infection in HCT recipients [14]. This finding was supported by the fact that upregulation of the KIR expression of 2DS2 and 2DS4 protected against CMV reactivation in the early post-HCT time. *KIR2DS2* and *KIR2DS4* expression was elevated in individuals after HCT compared with donor expression prior to transplant and in recipients with CMV DNAemia compared with non-viremic recipients [15]. In another study of a D+/R− organ transplant recipient cohort, the presence of *KIR2DS2* in combination with *KIR2DL3* was associated with protection against CMV viremia. However, this protective effect was only present if neither the donor nor the recipient expressed any HLA-C2 molecules [16]. In immunocompetent individuals, *KIR2DS2* was found to be negatively associated with the risk of infection, in which it was more frequent in controls than in those with CMV infection (OR = 0.14) [17]. Additionally, the binding of *KIR2DS2* to different peptides that derive from diverse pathogens and to different HLA class I molecules is particularly interesting given its protective role in acute viral encephalitis and chronic hepatitis C virus infection [18,19,20]. Interactions between viral epitopes, KIRs that are expressed on NK cells, and HLA ligands can lead to either the activation or inhibition of cytotoxic NK cell activity. KIR and HLA genotypes segregate independently of each other. Therefore, a proportion of KIRs likely has no HLA ligand. Conversely, individuals may possess HLA ligands for which they have no KIR. Group 1 HLA-C (HLA-C1) allotypes are characterized by an asparagine residue at position 80. They are ligands for the inhibitory receptors *KIR2DL2* and *KIR2DL3*, which segregate as alleles of a single locus. *KIR2DL2* binds HLA-C1 with greater affinity than *KIR2DL3* [21]. Our results showed that subjects who had *KIR2DL3* and *KIR2DL2*–HLA-C1 were more likely to develop CMV infection. These findings are consistent with previous reports of KIR and HLA genotyping, indicating that the absence of HLA ligands for inhibitory KIRs could be associated with a significant reduction of CMV infection [11]. The cumulative evidence shows that *KIR2DL2*–HLA-C1 and *KIR2DL3* may be molecular biomarkers of virus-induced diseases [22,23,24].

To date, it has been difficult to demonstrate the binding of *KIR2DS2* to HLA-C1. The sequences of *KIR2DS2* are very similar to *KIR2DL2.* Despite shared sequence homology, one key difference between *KIR2DS2* and *KIR2DL2* is the presence of a tyrosine (compared with a phenylalanine) residue at position 45, and this is thought to substantially affect the binding of *KIR2DS2* to HLA-C [25]. Steward et al. examined the binding of activating KIR to cells that were infected with human herpes viruses, including CMV, inducing no detectable *KIR2DS2* ligand at the cell surface [26]. A very low affinity of the *KIR2DS2*–HLA-C interaction that does not have the same degree of peptide selectivity as other KIR–HLA interactions was the reason that we did not analyze *KIR2DS2*–HLA-C. We are aware that the lack of information concerning KIR2DS2 ligands may be a limitation of the present study.

Other authors did not find that KIR and HLA genotypes are related to protection against CMV infection in kidney transplant recipients [27]. One possible explanation for this finding is that the previous study analyzed a patient cohort with a high risk of CMV infection (i.e., exclusively CMV-negative recipients of a CMV-positive donor kidney). *KIR2DL2* and *KIR2DS2* belong to the KIR B haplotype. The majority (74.8%) of our recipients were CMV-seropositive before transplantation. Evidence indicates that KIR B haplotypes modify the risk of CMV viremia solely in patients who have already been exposed to CMV before organ transplantation, suggesting that “primed” or “memory-like” NK cells are cellular correlates for this protective effect [28].

The incidence of CMV infection or reactivation was proposed to be increased by the heritance of multiple activating KIR genes [12,29]. We observed a trend toward a lower incidence of CMV infection in recipients who carried the KIR B/X genotype, but no significant difference was found between both the KIR A/A and KIR B/X genotypes (*p* = 0.065). The degree of protection against CMV was also suggested to increase with the number of activating KIR genes [12]. We analyzed the potential additive protective effect of the number of activating KIRs relative to CMV occurrence, but we did not confirm such an association.

In the present study, a protective effect against CMV infection in recipients who had the *KIR2DS2* genotype was found in 73.6% of the individuals, but 26.4% who had this protective genotype still had CMV infection. This may be attributable to the fact that the KIR genotype profile does not necessarily correlate with concurrent expression. The expression of *KIR2DS2* was previously shown to vary among genotype-positive healthy individuals, in which some were expressers and some were non-expressers [15]. We did not investigate the expression of KIRs, which may be considered a limitation of our study. 

Similar to previous reports, the present study confirmed that donor and recipient (D/R) CMV serostatus and allograft dysfunction were key predictors of the risk of CMV infection after kidney transplantation [2,3,5,30]. Renal insufficiency is associated with immune dysfunction, including lymphocytopenia, which contributes to the high prevalence of infections [31]. The present study confirmed our previous findings that total lymphocyte count before posttransplant day 90 is a risk factor for posttransplant CMV infection [32]. This phenomenon has also been described in liver transplant patients [33,34].

Interestingly, in the present study, an earlier time of antiviral prophylaxis initiation was not significant in the univariate analysis, but it became a significant predictor of posttransplant CMV infection in the multivariate analysis. Antiviral prophylaxis that is initiated promptly after transplantation could impair the development of an adequate CMV-specific response of cytotoxic and helper T lymphocytes, which seems to be crucial for protection against CMV infection. This finding is consistent with another study that found that a 14-day delay in the initiation of long-term prophylaxis in (D+/R−) organ transplant recipients could prevent the development of late CMV infection [35].

## 4. Materials and Methods

### 4.1. Study Design, Participants, and Outcome Parameters

Patients who received a kidney transplant between March 2007 and November 2014 in the Department of General and Transplant Surgery, T. Orłowski Institute, were enrolled in the study (*n* = 138). Deceased donor grafts were typed for HLA A, B, and DR. Kidney transplant recipients were typed for KIR and HLA A, B, C, and DR. Afterward, close monitoring of the occurrence of CMV infection, leucopenia, lymphocytopenia, and acute rejection episodes was performed for 720 days posttransplantation according to following scheme: every seven days in the first month, every two weeks in months 2 and 3, every four weeks in months 4, 5, and 6, and every three months later on. Additionally, CMV DNA-emia and kidney protocol biopsies were performed at the end of the third posttransplant month. Each case the clinical suspicion of CMV infection was verified by plasma CMV DNAemia. Recipients suspected of graft rejection were subjected to kidney biopsy. The eGFR was calculated by the abbreviated Modification of Diet in Renal Disease equation. A total of 61.6% of the recipients received induction therapy, and 87.7% received triple maintenance therapy with a combination of prednisone, tacrolimus, and antimetabolite (mycophenolatemofetil or mycophenolate sodium; Table 1). According to the institutional protocol, all of the patients received antiviral prophylaxis. A total of 83% of the recipients received valganciclovir, with the dose adjusted to kidney graft function, and 17% were given acyclovir (600 mg/day) for at least 90 days postoperatively. Based on the criteria that were recommended by the American Society of Transplantation for use in clinical trials, the following definitions applied:CMV infection (CMV DNAemia regardless of symptoms) and CMV disease (evidence of CMV infection with attributable symptoms) [36]. All rejection episodes were proven by biopsy, with the diagnosis based on Banff 2017criteria [37]. T-cell-mediated rejection was initially treated with 500 mg methyloprednisolone intravenously for three consecutive days; if resistance to this treatment was evident, then the patients were treated with thymoglobulin. Antibody-mediated rejection episodes were treated with intravenous methyloprednisolone, immunoglobulin, and/or plasmapheresis.

The study was approved by the Ethical Committee of the Medical University of Warsaw (KB/114/2014) and complied with the provisions of Good Clinical Practice Guidelines and the Declaration of Helsinki. All of the patients provided informed consent prior to participation in the study.

### 4.2. Procedures

#### 4.2.1. KIR and HLA Genotyping

Donors and recipients were typed for HLA-A, HLA-B, and HLA-DR-B1 by sequence-specific primer polymerase chain reaction (PCR). Peripheral blood specimens were collected, and mononuclear cell samples were isolated. DNA was extracted and stored at –80 C until use. DNA concentrations of each sample were determined using a UV–VIS NanoDrop Spectrophotometer (ThermoFisher Scientific, Wilmington, DE, USA). All further genotyping was performed retrospectively (i.e., at the end of follow up). DNA from mononuclear cells was used for KIR genotyping by applying a reverse sequence specific oligonucleotide (SSO) method according to the manufacturer’s instructions (Lifecodes KIR Genotyping, Immucor Transplant Diagnostic, Stamford, CT, USA). The HLA-C and KIR genes were typed using Lifecodes HLA-SSO and KIR-SSO typing kits, respectively (ImmucorDgn, Lifecodes, Stamford, CT, USA) based on xMAP Luminex technology (Luminex, Austin, TX, USA) according to the manufacturer’s instructions. The presence or absence of the following 16 KIR genes was identified: *KIR2DL1*, *KIR2DL2*, *KIR2DL3*, *KIR2DL4*, *KIR2DL5*, *KIR2DS1*, *KIR2DS2*, *KIR2DS3*, *KIR2DS4*, *KIR2DS5*, *KIR3DL1*, *KIR3DL2*, *KIR3DL3*, *KIR3DS1*, *KIR2DP1*, and *KIR3DP1*. KIR gene profiles were determined by the presence or absence of each KIR gene in a given individual. This method of KIR typing does not allow the direct determination of *KIR2DL2* copy number. Instead, we used the allelic nature of *KIR2DL2* and *KIR2DL3* at the 2DL2/2DL3 locus to infer the number of copies of *KIR2DL2*. *KIR2DS4* was typed for encoded cell-surface receptor (full) or a truncated protein variant with loss of the transmembrane and cytoplasmic domains (del) by GeneScan analysis with a 6-FAM labeled primer. The deleted variant of *KIR2DS4* was not anchored to the cell membrane but was encoded for a soluble form of the protein that is potentially secreted and likely lacks function(s). We used the MATCH IT DNA program for analysis.

The presence of KIR ligands in recipients was assessed by grouping each patient’s HLA class I antigens according to defined specificities: patients were considered to have C1 group ligands if they possessed an HLA-C molecule with an asparagine at position 80 (e.g., HLA-Cw1, -Cw3, -Cw7, and -Cw8), C2 group ligands if they possessed an HLA-C molecule with a lysine residue at position 80 (e.g., HLA-Cw2, -Cw4, -Cw5, and -Cw6), and Bw4 ligands if their HLA-B antigens included at least one antigen with Bw4 specificity (e.g., HLA-B5, -B13, -B17, and -B27) as described elsewhere [13,38,39]. We defined KIR-A and KIR-B haplotypes as previously described. In brief, KIR-A haplotypes only contain *KIR2DL1*, *KIR3DL1*, and *KIR2DS4* and four framework genes (*KIR2DL4*, *KIR3DL2*, *KIR3DL3*, and *KIR2DL3*). KIR-B haplotypes contain one or more of the following genes: *KIR2DL2*, *KIR2DL5*, *KIR2DS1*, *KIR2DS2*, *KIR2DS3*, *KIR2DS5*, and *KIR3DS1*.

#### 4.2.2. CMV DNAemia

Cytomegalovirus DNAemia was evaluated in plasma using a commercial quantitative nucleic acid amplification test (SmartCycler II, Cepheid AB, Sunnyvale, CA, USA) that from 2013 was calibrated to the 1st World Health Organization International Standard. The limit of detection was 50 copies/mL, with a linearity of 500–107 copies/mL.

#### 4.2.3. Detection of CMV-Specific Antibodies

Specific serum anti-CMV IgG titers were measured using the ARCHITECT CMV IgG assay (Abbott Laboratories, Dublin, Ireland) according to the manufacturer’s instructions. The ARCHITECT CMV IgG assay is a chemiluminescent microparticle immunoassay that is designed to have a precision of ≤10% total (total is the accumulation of within run, between run, and between day) coefficient of variation (CV) for representative specimens within the ranges of 6–60 AU/mL and 200–250 AU/mL. Among transplant recipients, relative sensitivity is 100% (lower 95% confidence limit of 91.96%), and specificity is 100% (lower 95% confidence limit of 93.62%; from the manufacturer’s information brochure).

### 4.3. Statistical Analyses

Qualitative variables were compared using the *χ*^2^ test and Fisher’s exact test, respectively, to the sample size. To measure the fraction of events, the Kaplan-Meier model was used. This model estimates and tests survival over time under different factors. Quantitative variables were summarized by medians (ranges) because the parameters did not follow a normal distribution, and they were compared using the Wilcoxon Rank-Sum test. Multidimensional analysis was performed with the generalized logistic regression model (GLM). To assess the goodness-of-fit and select the optimal model, the Akaike Information Criterion statistic was used. Values of *p* < 0.05 were considered statistically significant. The data were analyzed using SAS/STAT 14.3 [SAS Institute Inc., Cary, NC, USA] software.

## 5. Conclusions

In summary, our results suggest that genotyping recipients for the *KIR2DS2*, *KIR2DL2*, and *KIR2DL3* genes may help predict the occurrence of posttransplant CMV infection. Other factors, such as graft function, time of antiviral prophylaxis initiation, lymphocyte blood count, and pretransplant serostatus, were independent predictive factors for the occurrence of CMV. Currently, the risk stratification for posttransplant CMV infection is primarily based on the assessment of D/R CMV serostatus prior to transplantation. In our opinion, this strategy could be improved by implementing other biomarkers that are predictive of CMV infection after kidney transplantation, which would allow for more personalized management. The assessment of *KIR2DS2* prior to transplantation could potentially be used in combination with D/R serostatus to more accurately predict the risk of CMV infection, including the precise identification of transplant individuals who require a longer duration of antiviral prophylaxis therapy in individuals who do not have *KIR2DS2*. In conclusion, our findings confirm that KIR/HLA genotypes play a significant role in anti-CMV immunity and suggest the contribution of not only environmental but also genetic factors in the incidence of CMV infection after kidney transplantation.

## Figures and Tables

**Figure 1 ijms-20-00546-f001:**
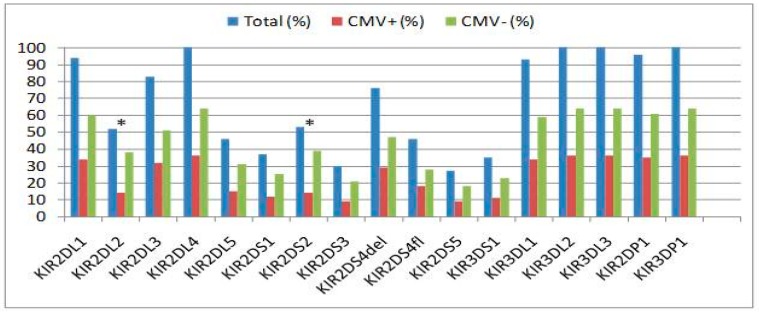
Frequency of CMV infection (CMV+) relative to each killer cell immunoglobulin-like receptor gene. The asterisk (*) indicates statistical significance.

**Figure 2 ijms-20-00546-f002:**
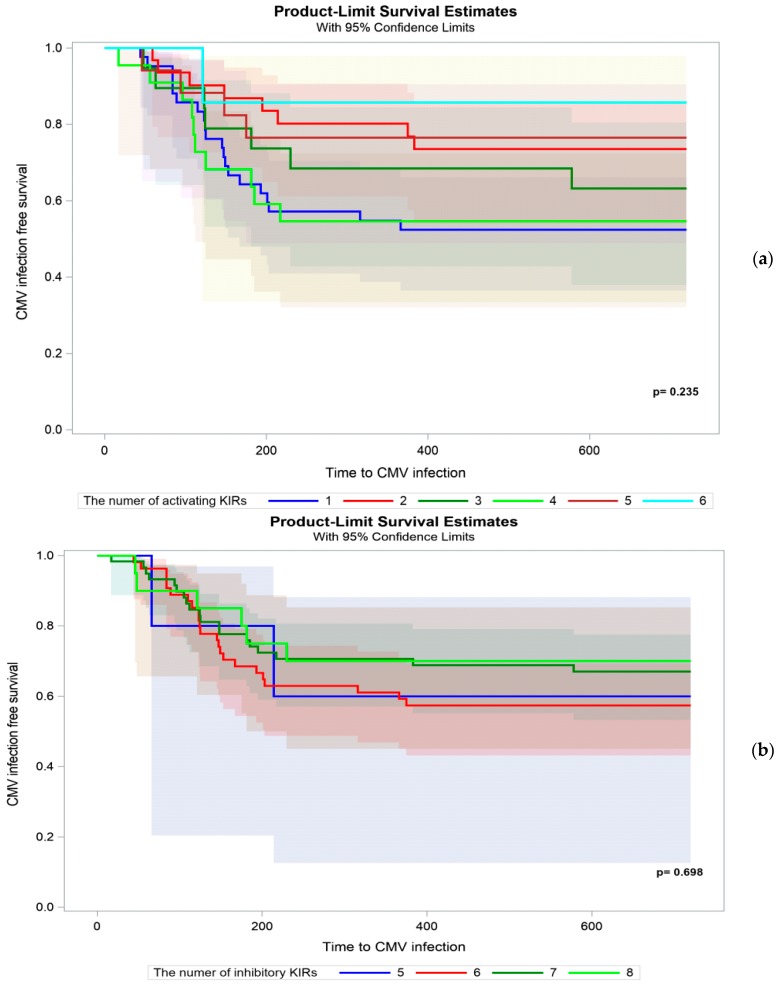
Occurrence of CMV infection in kidney transplant recipients stratified by the cumulative number of (**a**) activating and (**b**) inhibitory KIR genes.

**Figure 3 ijms-20-00546-f003:**
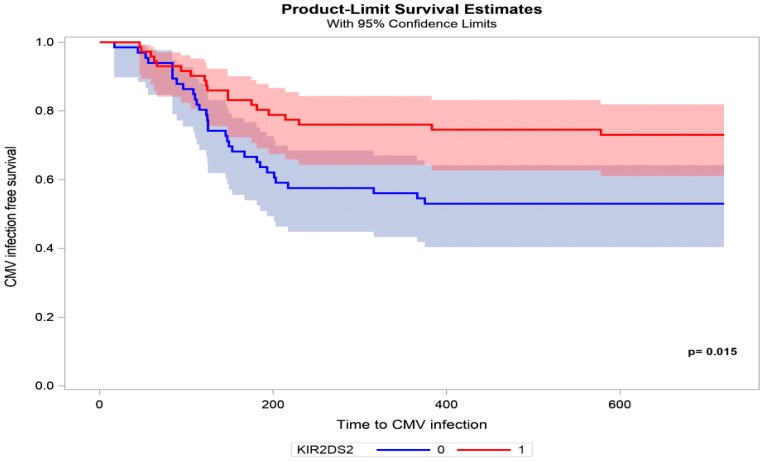
Time from kidney transplantation to CMV infection in patients with (red line) vs. without (blue line) KIR2DS2.

**Table 1 ijms-20-00546-t001:** Baseline characteristics of the study patients.

Characteristic	
Age of recipient (years) (median (range))	48 (20–77)
Gender (male)	*n* = 90 (65%)
**Type of Transplant**	
Kidney	*n* = 133 (96.4%)
Kidney + pancreas	*n* = 4 (2.9%)
Kidney + heart	*n* = 1 (0.7%)
**Type of Donor**	
Living	*n* = 10 (7.2%)
Deceased	*n* = 128 (92.8%)
**Pretransplant Donor (D)/Recipient (R) CMV Serostatus**	
D+/R−	*n* = 34 (25.2%)
D− or D+/R+	*n* = 104 (74.8%)
**Induction Therapy**	
Thymoglobulin	*n* = 13 (9.4%)
Basiliximab	*n* = 72 (52.2%)
None	*n* = 53 (38.4%)
**Maintenance Immunosuppression**	
tacrolimus + mycophenolate mofetil/sodium + prednisone	*n* = 121 (87.7%)
cyclosporine A + mycophenolate mofetil/sodium + prednisone	*n* = 15 (10.9%)
tacrolimus + everolimus + prednisone	*n* = 2 (1.4%)
Time from kidney transplant to antiviral prophylaxis initiation (days) (mean ± SD [range])	6 ± 5 (0–25)
Time from kidney transplant to antiviral prophylaxis discontinuation (days) (mean ± SD [range])	90 ± 21 (12–178)
Duration of antiviral prophylaxis (days) (mean ± SD (range))	84 ± 21(10–175)
Allograft function (eGFR; ml/min/1.73 m^2^) (mean ± SD [range])	
Day 30	46.7 ± 19.9 (6.7–103.7)
Day 90	47.8 ± 18.5 (8.5–98.9)
Day 360	49.5 ± 18.6 (8.8–105.0)
**KIR Genotype**	
A/A	*n* = 42 (30.4%)
B/X	*n* = 96 (69.6%)

eGFR, estimated glomerular filtration rate; SD, standard deviation.

**Table 2 ijms-20-00546-t002:** Frequency of each KIR gene in 138 patients.

	KIR Genes												
	3DS1	2DL1	2DL2	2DL3	2DL5	2DS1	2DS2	2DS3	2DS4fl	2DS4del	2DS5	3DL1	2DP1
*N*	48	130	71	115	64	51	72	42	63	106	37	128	132
%	34.8	94.2	51.4	83.3	46.4	36.9	52.2	30.4	45.7	76.8	26.8	92.8	95.7

**Table 3 ijms-20-00546-t003:** Association between KIRs and KIR genotypes and CMV infection.

Genetic factor	CMV Infection *n* (%)	No CMV Infection *n* (%)	*p*
**KIR Genotype**			0.065
A/A	20 (47.6)	22 (52.4)	
B/X	30 (31.2)	66 (68.8)	
**Number of Activating KIRs**			0.201
1	20 (47.6)	22 (52.4)	
2	8 (25.8)	23 (74.2)	
3	7 (36.8)	12 (63.2)	
4	10 (45.5)	12 (54.5)	
5	4 (23.5)	13 (76.5)	
6	1 (14.3)	6 (85.7)	
**Number of Inhibitory KIRs**			0.631
5	2 (40.0)	3 (60.0)	
6	23 (42.6)	31 (57.4)	
7	19 (32.2)	40 (67.8)	
8	6 (30.0)	14 (70.0)	
KIR2DL1–HLA-C2	29 (37.2)	49 (62.8)	0.791
KIR2DL2–HLA-C1	17 (30.9)	38 (69.1)	0.289
KIR2DL3–HLA-C1	34 (40.0)	51 (60.0)	0.243
KIR3DL1–HLA-Bw4	35 (37.2)	59 (62.8)	0.720
KIR3DL2–HLA-A3/A11	22 (38.6)	35 (61.4)	0.627

KIR, killer-cell immunoglobulin-like receptor; HLA, human leukocyte antigen.

**Table 4 ijms-20-00546-t004:** Clinical characteristics of kidney transplant recipients stratified by the occurrence of CMV infection.

Characteristic	CMV Infection	No CMV Infection	*p*
Age (years) (mean ± SD)	50.3 ± 13.6	46.5 ± 14.6	0.143
Gender, male (n [%])	30 (60.0)	60 (68.2)	0.332
**Type of Transplant (n [%])**			
Kidney	48 (36.1)	85 (63.9)	0.371
Kidney + pancreas	1 (25)	3 (75)	
Kidney + heart	1 (100)	0	
Donor (deceased/living) (n [%])	49 (38.3)/1(10)	79 (61.7)/9 (90)	0.073
**Allograft Function (eGFR) (mean ± SD)**			
Day 30	41.6 ± 16.4	49.6 ± 21.1	0.040
Day 90	40.9 ± 17.1	51.6 ± 18.2	0.0006
**Acute Rejection (n [%])**			
TCMR	18 (42.9)	24 (57.1)	0.639
ABMR	8 (47.1)	9 (52.9)	0.521
TCMR or ABMR	22 (44.9)	27 (55.1)	0.362
**HLA Mismatch (n [%])**			
≤3/6	25 (40.3)	37 (59.7)	0.366
>3/6	25 (32.9)	51 (67.1)	
**Lymphocyte blood count ≤ day 90 (G/L) (n [%])**			
<0.8	31 (56.4)	24 (43.6)	0.0003
≥0.8	17 (24.6)	52 (75.4)	
**Leukocyte Blood Count ≤ Day 90 (G/L) (n [%])**			
<4.1	34 (46.6)	39 (53.4)	0.007
≥4.1	16 (24.6)	49 (75.4)	
**Donor (n [%])**			
CMV IgG+	45 (37.5)	75 (62.5)	0.752
CMV IgG−	5 (33.3)	10 (66.7)	
**Pretransplant Recipient CMV Serostatus (n [%])**			
CMV IgG+	32 (31.7)	69 (68.3)	0.066
CMV IgG−	18 (48.7)	19 (51.3)	
Pretransplant recipient CMV IgG titer day 0 (AU/mL) (mean ± SD)	80.7 (95.9)	133.4 (116.6)	0.016
**Pretransplant Donor/Recipient CMV Serostatus (n [%])**			
D+/R−	18 (52.9)	16 (47.1)	0.019
D+ or D−/R+	32 (30.8)	72 (69.2)	
Duration of antiviral prophylaxis (days) (mean ± SD)	80 ± 23	87 ± 19	0.276
Time from kidney transplant to antiviral prophylaxis initiation (days) (mean ± SD)	4.68 ± 4.97	7.22 ± 5.39	0.005
**Induction Therapy (%)**			
Yes (basiliximab)	27.8	72.2	0.076
Yes (thymoglobulin)	53.8	46.2	
No	43.4	56.6	
**Induction Therapy (%)**			
Yes (basiliximab or thymoglobulin)	31.8	68.2	0.166
No	43.4	56.6	

eGFR, estimated glomerular filtration rate; TCMR, T-cell mediated rejection; ABMR, antibody-mediated rejection; HLA, human leukocyte antigen; SD, standard deviation; AU, arbitrary units.

**Table 5 ijms-20-00546-t005:** Best subset of variables with the strongest predictive value for the incidence of posttransplant CMV infection according to the multivariate analysis.

Best Subset of Parameters	Multivariate Analysis		
	OR	95% CI	*p*
Lack of KIR2DS2	7.984	1.155–55.215	0.04
KIR2DL2-HLA-C1	8.197	1.055–62.500	0.04
KIR2DL3	4.219	0.866–20.833	0.08
Allograft dysfunction	1.030	1.002–1.059	0.04
Lymphocytopenia	3.200	1.286–7.960	0.01
Earlier time of antiviral prophylaxis initiation	1.107	1.013–1.212	0.03
D+/R− pretransplant serostatus	3.241	1.043–10.069	0.04

KIR, killer-cell immunoglobulin-like receptor; CI, confidence interval; OR, odds ratio; D, donor; R, recipient.

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
