# Peer review of "Killer Immunoglobulin-Like Receptor 2DS2 (KIR2DS2), KIR2DL2-HLA-C1, and KIR2DL3 as Genetic Markers for Stratifying the Risk of Cytomegalovirus Infection in Kidney Transplant Recipients"

_ijms, 2019, doi:10.3390/ijms20030546_

Round 1
Reviewer 1 Report
Manuscript ID: ijms-419671
Title: Killer Immunoglobulin-Like Receptor 2DS2 (KIR2DS2), KIR2DL2-HLA-C1, and KIR2DL3 as genetic markers for stratifying the risk of cytomegalovirus infection in kidney transplant recipients
In this manuscript authors investigated whether natural killer cell immunoglobulin-like receptors (KIRs) genes and their ligands affect the occurrence of cytomegalovirus (CMV) infection in a group of (n=138) kidney transplant recipients. By means of multivariate analysis and considering the KIR genes typing and the outcome of CMV posttransplant infection, authors came to the conclusion that the KIR/HLA genotype plays a significant role in anti-CMV immunity and suggest the contribution of both environmental and genetic factors to the incidence of CMV infection after kidney transplantatation.
This is an interesting study adding a piece to the puzzle of CMV posttransplantation infection, a significant cause of serious posttransplant complications. Authors managed to include a large set of samples (N=138) and use a robust methodological approach, both contributing to the strength of their conclusions. I really do not have much to say negatively and advise publication. I do have one question, if the Banff 2017 criteria were set and published in February 2018 (see: Haas et al. The Banff 2017 Kidney Meeting Report: Revised diagnostic criteria for chronic active T cell-mediated rejection, antibody-mediated rejection, and prospects for integrative endpoints for next-generation clinical trials. Am J Transplant. 2018 Feb;18(2):293-307), how did authors follow these criteria on proving rejections by biopsy if rejections have occurred between 2007 and 2014? Please explain
Author Response
Response to Reviewer 1 Comments
Point 1.This is an interesting study adding a piece to the puzzle of CMV posttransplantation infection, a significant cause of serious posttransplant complications. Authors managed to include a large set of samples (N=138) and use a robust methodological approach, both contributing to the strength of their conclusions. I really do not have much to say negatively and advise publication. I do have one question, if the Banff 2017 criteria were set and published in February 2018 (see: Haas et al. The Banff 2017 Kidney Meeting Report: Revised diagnostic criteria for chronic active T cell-mediated rejection, antibody-mediated rejection, and prospects for integrative endpoints for next-generation clinical trials. Am J Transplant. 2018 Feb;18(2):293-307), how did authors follow these criteria on proving rejections by biopsy if rejections have occurred between 2007 and 2014? Please explain.
Response 1 : We appreciate this comment. The morphology of kidney biopsies has been interpreted according to subsequent editions of Banff classification (meeting reports 2009, 2011, 2013, 2015). We replaced the previous reference with a one that reviews all historical and current editions of Banff classification:
38. Roufosse C, Simmonds N, Clahsen-van Groningen M, Haas M, Henriksen KJ, Horsfield C, Loupy A, Mengel M, Perkowska-Ptasińska A, Rabant M, Racusen LC, Solez K, Becker JU. A 2018 Reference Guide to the Banff Classification of Renal Allograft Pathology. Transplantation. 2018 Nov;102(11):1795-1814.
Reviewer 2 Report
In this study, authors tested KIR genes and ligands in 138 kidney transplant recipients, and associated with occurrence of CMV infection. It seems to be interesting but, the methods and results are not clear.
1. The authors should present the association of CMV infection with individual KIR gene, KIR ligand, KIR ligand combinations or KIR haplotypes by univariate analysis.
2. The authors should provide a detailed description of CMV surveillance.
3. Please provide a definition or reference of CMV infection and disease.
4. To present the effect of KIR gene on CMV reactivation, cumulative incidence of CMV infection in recipients should be analyzed (x-axis; days post-transplantation, Y-axis; incidence of CMV infection) between patients with and without specific KIR gene (including statistical significance).
5. Statistical comparison of KIR gene numbers (Figure 2) should be given.
Author Response
Response to Reviewer 2 Comments
Point 1. The authors should present the association of CMV infection with individual KIR gene, KIR ligand, KIR ligand combinations or KIR haplotypes by univariate analysis .
Response 1: Thank you for this comment. The original manuscript contains the results of univariate analysis (paragraph 2.). We decided to change the sequence of paragraphs 2.2 and 2.3 and the title of the paragraph 2.2 hoping that it will make the contents of his section of the manuscript more clear.
A new title of paragraph 2.2. is: The relation between the CMV infection occurrence and individual KIR gene frequencies (univariate analysis).
Point 2. The authors should provide a detailed description of CMV surveillance.
Response 2: We appreciate this comment. Close monitoring of the occurrence of CMV infection, leucopenia, lymphocytopenia, and acute rejection episodes was performed for 720 days posttransplantation according to following scheme: every 7 days in the first month, every 2 weeks in months 2. and 3., every 4 weeks in months 4., 5., 6, and every 3 months later on. Additionally CMV DNA-emia and kidney protocol biopsies were performed at the end of 3rd posttransplant month. Each case the clinical suspicion of CMV infection was verified by plasma CMV DNAemia. Recipients suspected of graft rejection were subjected to kidney biopsy. We added this information in the revised manuscript.
Point 3. Please provide a definition or reference of CMV infection and disease.
Response 3: The definition (plus the reference) of CMV infection and CMV disease is presented in lines 256-259 of the original manuscript: "Based on the criteria that were recommended by the American Society of Transplantation for use in clinical trials, the following definitions applied: CMV infection (CMV DNAemia regardless of symptoms) and CMV disease (evidence of CMV infection with attributable symptoms)[37]."
If the Reviewer wishes we will expand the given definition.
Point 4. To present the effect of KIR gene on CMV reactivation, cumulative incidence of CMV infection in recipients should be analyzed (x-axis; days post-transplantation, Y-axis; incidence of CMV infection) between patients with and without specific KIR gene (including statistical significance).
Response 4: Free survival approach by the Kaplan Meier model is an equivalent to cumulative incidence approach [Breslov N., Day N., “Statistical Methods in Cancer Research Volume I: The Analysis of Case-Control Studies”, IARC Scientific Publication No. 32, 1980, chap. II]. The results of the models were presented in figures 2. and 3. The significance of the differences in KIR genes status were examined using the log-rank test.
Point 5. Statistical comparison of KIR gene numbers (Figure 2) should be given.
Response 5: We revised the manuscript accordingly.